# Good recovery of immunization stress-related responses presenting as a cluster of stroke-like events following CoronaVac and ChAdOx1 vaccinations

**Metha Apiwattanakul**[1,2]*, **Narupat Suanprasert**[1], **Arada Rojana-Udomsart**[1],
**Thanes Termglinchan**[1], **Chaichana Sinthuwong**[1], **Tasanee Tantirittisak**[1],
**Suchat Hanchaiphiboolkul**[1], **Pantep Angchaisuksiri**[2,3], **Suphot Srimahachota**[2,4],
**Jurai Wongsawat**[2,5], **Somjit Stiudomkajorn**[2,6], **Sasisopin Kiertiburanakul**[2,3],
**Chonnamet Techasaensiri**[2,7], **Wannada Laisuan**[2,8], **Weerawat Manosuthi**[2,5],
**Pawinee Doungngern**[2,9], **Wereyarmarst Jaroenkunathum**[2,10],
**Teeranart Jivapaisarnpong**[2,11], **Apinya Panjangampatthana**[2,9], **Jirapa Chimmanee**[2,9],
**Kulkanya Chokephaibulkit**[2,12,13]

1 Department of Neurology, Neurological Institute of Thailand, Ministry of Public Health, Nonthaburi, Thailand, 2 Department of Disease Control, the National AEFI Committee, Ministry of Public Health, Nonthaburi, Thailand, 3 Department of Medicine, Faculty of Medicine Ramathibodi Hospital, Mahidol University, Bangkok, Thailand, 4 Department of Medicine, Faculty of Medicine, King Chulalongkorn Memorial Hospital, Bangkok, Thailand, 5 Department of Diseases Control, Bamrasnaradura Infectious Diseases Institute, Ministry of Public Health, Nonthaburi, Thailand, 6 Division of Neurology, Department of Pediatrics, Queen Sirikit National Institute of Child Health, Ministry of Public Health, College of Medicine, Rangsit University, Bangkok, Thailand, 7 Division of Infectious Diseases, Department of Pediatrics, Faculty of Medicine Ramathibodi Hospital, Mahidol University, Bangkok, Thailand, 8 Division of Allergy Immunology and Rheumatology, Department of Medicine, Faculty of Medicine Ramathibodi Hospital, Mahidol University, Bangkok, Thailand, 9 Division of Epidemiology, Department of Disease Control, Ministry of Public Health, Nonthaburi, Thailand, 10 Department of Medical Sciences, Institute of Biological Products, Ministry of Public Health, Nonthaburi, Thailand, 11 National Biopharmaceutical Facility, King Mongkut's University of Technology Thonburi, Bangkok, Thailand, 12 Department of Pediatrics, Faculty of Medicine Siriraj Hospital, Mahidol University, Bangkok, Thailand, 13 Siriraj Institute of Clinical Research, Faculty of Medicine Siriraj Hospital, Mahidol University, Bangkok, Thailand

* apiwattanakul.metha@gmail.com

**Data Availability Statement:** All relevant data are within the manuscript and its Supporting Information files.

## Abstract

### Background

Immunization stress-related responses presenting as stroke-like symptoms could develop following COVID-19 vaccination. Therefore, this study aimed to describe the clinical characteristics of immunization stress-related responses causing stroke-like events following COVID-19 vaccination in Thailand.

### Methods

We conducted a retrospective study of the secondary data of reported adverse events after COVID-19 immunization that presented with neurologic manifestations. Between March 1 and July 31, 2021, we collected and analyzed the medical records of 221 patients diagnosed with stroke-like symptoms following immunization. Two majority types of vaccines were

**Funding:** This study is funded the National Research Council of Thailand Year 2021 grant No. 74/2564. The funders had no role in the design and conduct of the study; collection, management, analysis, and interpretation of the data; preparation, review, or approval of the manuscript; and the decision to submit the manuscript for publication.

**Competing interests:** The authors have declared that no competing interests exist.

used at the beginning of the vaccination campaign, including CoronaVac (Sinovac) or ChAdOx1 (AstraZeneca). Demographic and medical data included sex, age, vaccine type, sequence dose, time to event, laboratory data, and recovery status as defined by the modified Rankin score. The affected side was evaluated for associations with the injection site.

## Results

Overall, 221 patients were diagnosed with immunization stress-related responses (stroke-like symptoms) following CoronaVac (Sinovac) or ChAdOx1 (AstraZeneca) vaccinations. Most patients (83.7%) were women. The median (interquartile range) age of onset was 34 (28–42) years in patients receiving CoronaVac and 46 (33.5–60) years in those receiving ChAdOx1. The median interval between vaccination and symptom onset for each vaccine type was 60 (16–960) min and 30 (8.8–750) min, respectively. Sensory symptoms were the most common symptomology. Most patients (68.9%) developed symptoms on the left side of the body; 99.5% of the patients receiving CoronaVac and 100% of those receiving ChAdOx1 had a good outcome (modified Rankin scores $\leq 2$, indicating slight or no disability).

## Conclusions

Immunization stress-related responses presenting as stroke-like symptoms can develop after COVID-19 vaccination. Symptoms more likely to occur on the injection side are transient (i.e., without permanent pathological deficits). Public education and preparedness are important for administering successful COVID-19 vaccination programs.

## Introduction

Immunization stress-related responses (ISRR) are defined by the World Health Organization (WHO) as symptoms and signs of bodily responses to vaccination [1]. These events are not caused by side effects due to vaccine components. Instead, they arise from the stress reaction to the vaccination process/cascade, which could be precipitated or potentiated by the pain or other common side effects of the injections. The symptoms may develop immediately (within 5 minutes) following vaccination, such as vasovagal reaction, or later, such as non-epileptic seizures, abnormal movements after COVID-19 vaccination [2] or motor and sensory symptoms which demonstrate positive neurological signs (Hoover's sign) which were diagnosed of functional neurological disorders [3].

Since the initiation of the COVID-19 vaccination campaign in Thailand on February 28, 2021, 17,685,974 doses have been administered through the study follow-up date of July 31, 2021. Serious or concerning adverse events occurring within 30 days after the vaccination as well as event clusters were reported to the Adverse Event Following Immunization (AEFI) committee at the Department of Disease Control (Thailand Ministry of Public Health). All cases were reviewed by the AEFI committee. Causality and relationships to vaccination were determined by consensus.

The first ISRR cluster was identified at the start of April 2021; the committee identified five patients with stroke-like symptoms. These patients were healthcare workers (HCWs; i.e., the first group to receive the vaccine in the country). Similarly, within one week, we identified many clusters of stroke-like symptoms reported following vaccination, all of which occurred in HCWs.

This information spread rapidly through social networks and caused vaccine hesitancy. Impurity or high vaccine specificity was suspected as probable causes. However, the relevant vaccine types were investigated and found within regulated quality without identifiable toxins or contamination. The AEFI committee convened to review these cases in a timely fashion and found that most of the reported cases were those of ISRR. Thus, herein, we report on the clinical features of clustered ISRR presenting with neurologic manifestations to inform public health education, epidemic preparedness, and vaccine administration programs.

## Materials and methods

### Study design and participants

Thailand initiated a nationwide COVID-19 vaccination campaign on February 28, 2021. The AEFI report system for the routine vaccine in the National Immunization Program (NIP) has been established in every public hospital for over twenty years. The dedicated hospital staff reports any serious AE or AE of special interest or AE that occurred in the cluster through the national AEFI surveillance program (AEFI-DDC) hosted by the Department of Disease Control, Ministry of Public Health. A medical record and imaging scan copy would be submitted for systematic review by the AEFI committee. In some cases of doubt, the treating clinician was contacted for additional information. The AEFI committee convened promptly to determine causality as well as the relationships of the presenting symptomology with vaccination status and timing. The primary physicians may primarily diagnose ISRR; however, all were confirmed by the AEFI committee composed of neurologists, internists, pediatricians, cardiologists, hematologists, and allergists. In some cases of doubt, the committee would seek further external expert neurologist consultation.

We conducted a retrospective study using the data received from the National AEFI surveillance program (AEFI-DDC) and the copy of the medical records of the cases that were determined to be ISRR by the AEFI committee. We requested only the specific data relevant to the ISRR from the Department of Disease Control, Ministry of Public Health. Specifically, we evaluated ISRR with a neurologic presentation that occurred between March 1 and July 31, 2021. The data evaluated in this secondary analysis were obtained without direct contact with patients or their attending healthcare workers. Neurological ISRRs were diagnosed based on the presence of clinical neurological symptoms compatible with dissociative neurological symptom reactions [1].

The data used for this analysis, including sex, age, vaccine type, sequence dose, time to event, clinical manifestations, laboratory data, and recovery status (which was defined according to the patients' modified Rankin scale [mRS] scores, with higher scores indicating a greater level of disability) were abstracted from the study database provided by the AEFI committee. The mRs scores were selected as the primary outcome measure for defining recovery status. A good outcome was defined as an mRS score of 0–2 (indicating slight or no disability), a poor outcome as an mRS score of 3–5 (indicating moderate to severe disability), and mortality was categorized as an mRS score of 6.

This study was approved by the Ethical Review Committee for Research in Human Subjects at the Thailand Ministry of Public Health approval No. 15/2564. The requirement for informed consent was waived due to the retrospective nature of this study and the fact that we conducted a secondary analysis of anonymized and de-identified data.

### Statistical analysis

When evaluating baseline demographic and medical data, we report medians and interquartile ranges (IQR) for continuous data (some variables were not normally distributed), whereas

categorical variables are presented as counts and percentages. Demographic and medical data were dichotomized by vaccine type (CoronaVac [Sinovac Biotech, Beijing, China], ChAdOx1 [AstraZeneca, Cambridge, UK]). Also, we evaluated the distributions and characteristics of adverse events by injection site. All statistical analyses were performed using GraphPad Prism 9 software (San Diego, CA, USA).

## Results

Following a total of 17,685,974 COVID-19 vaccine doses that were administered as of July 31, 2021 (including 8,699,803 doses of the CoronaVac vaccine, 8,000,079 doses of the ChAdOx1 vaccine and 986,092 doses of the BBIBP-CorV), 293 patients with severe neurological complications were reported to the AEFI committee. These included 278 and 15 patients who had received the CoronaVac and ChAdOx1 vaccines, respectively.

ISRR was diagnosed in 263 patients. Other neurological diseases were identified in 18 patients, including five patients with stroke (two with ischemic stroke and three with hemorrhagic stroke), four patients with neuropathy, four with provoked seizure, one with severe headache, one with severe myalgia, one with facial edema and numbness, one experiencing syncope, and one with aggravated back pain due to spinal stenosis. For 12 patients who reported severe neurological symptoms, the AEFI committee determined the causality as inconclusive due to incomplete medical records. Of the 263 patients reported to have ISRR, 221 had adequate data available for this secondary analysis. All patients had normal findings on the brain CT computed tomography (CT) and magnetic resonance imaging (MRI) scans. Table 1 shows demographic data, clinical symptomology, and health outcomes in the identified patients experiencing adverse vaccine-associated events.

Most of these patients were females (185/221, 87%); the median age of onset was 34 years (IQR, 28–42) for patients who received the CoronaVac vaccine and 46 years (33.5–60) for those who received the ChAdOx1 vaccine. The median interval between vaccination and symptom onset was 60 min (IQR, 16–960) for the CoronaVac vaccine and 30 min (8.8–750) for the ChAdOX1 vaccine. Most of these adverse events (175/221, 79.2%) occurred following

**Table 1. Clinical and demographic data for immunization stress-related responses (ISRR).**

| | CoronaVac vaccine (n = 211) | ChAdOX1 vaccine (n = 10) |
|---|---|---|
| Sex | | |
| Female (%) | 177 (84) | 8 (80) |
| Male (%) | 34 (16) | 2 (20) |
| Median age of onset (years, IQR) | 34, 28–42 | 46, 33.5–60 |
| Mean BMI (SD) | 24.7, 4.8 | 24.8, 4.5 |
| Median time from vaccination to symptom onset (min, IQR) | 60, 16–960 | 30, 8.8–750 |
| Number of patients following each dose | | |
| 1st dose | 165 | 10 |
| 2nd dose | 46 | - |
| Injection side | | |
| Left | 50 | 7 |
| Right | 4 | 0 |
| Unknown | 157 | 3 |
| Median time to recovery (days, IQR) | 2, 1–4 | 1, 4 hours—11 days |

IQR, interquartile range; SD, standard deviation

**Table 2. Clinical characteristic with respect to neurological symptoms stratified by the vaccine injection site.**

| | Left side injection | Right side injection | Unknown |
| --- | --- | --- | --- |
| | (n = 57) | (n = 4) | (n = 160) |
| Lateralization of symptoms | | | |
| Left side | 42 | 1 | 101 |
| Right side | 4 | 3 | 29 |
| Bilateral | 6 | - | 14 |
| Unknown | 2 | - | 7 |
| Non-stroke-like | 3 | - | 9 |
| Sensory symptoms | | | |
| Sensory (NA) | 2 | - | 7 |
| Sensory, left side | 20 | 1 | 59 |
| Sensory, right side | 1 | 2 | 19 |
| Sensory, bilateral | 2 | - | 5 |
| Motor symptoms | | | |
| Motor (NA) | - | - | - |
| Motor, left | 2 | - | 5 |
| Motor, right | 1 | 1 | 2 |
| Motor, bilateral | 1 | - | 5 |
| Sensorimotor (SM) symptoms | | | |
| SM (NA) | - | - | - |
| SM, left | 20 | - | 37 |
| SM, right | 2 | - | 8 |
| SM, bilateral | 3 | - | 4 |

NA, not applicable

the first vaccination dose. The median time from injection to recovery was 2 days (IQR, 1–4) and 1 day (IQR, 4 h to 11 days) for CoronaVac and ChAdOX1 recipients, respectively.

A total of 209 patients (94.5%) had sensory, motor, or combined sensorimotor symptoms. Of these, 144 (68.9%) patients developed symptoms on the left side of the body. Of the 61 patients whose injection side was known, 45 (73.8%) developed symptoms on the same side as the injection site. Table 2 shows clinical characteristics regarding neurological symptoms stratified by the vaccine injection site. In CoronaVac recipients (211 patients), 74.8% were fully recovered, and 99.5% had good outcomes (mRS 0–2). Only one patient had an mRS score of 3. This patient had clinical symptoms and laboratory data compatible with ISRR; the causes of the symptom recurrence and persistent high mRS score, which was inconsistent and likely to be functional deficits, are explored by a multidisciplinary team, including the psychiatrist. All 10 ChAdOX1 recipients had good outcomes (mRS 0–2). The mean BMI of participants with information on this variable (214 patients) was 24.67 kg/m$^2$ (standard deviation, 4.78). None of the patients in this study had ISRR after both doses of vaccines.

## Discussion

Several well-recognized neurological complications are associated with many forms of vaccination, including Guillain-Barré syndrome and acute disseminated encephalomyelitis. The immunological mimicry can explain the effects of vaccine antigens on the relevant myelin protein. However, stroke has not been recognized as an AEFI.; there are several reports of non-epileptic seizure, abnormal movement [2] or motor/sensory symptoms like stroke [3] from

COVID19 vaccination. Moreover, to the best of our knowledge, stroke-like syndromes occurring as clustered events have not been reported as adverse vaccine-associated events to date.

ISRR is considered an alternative term for functional neurological disorders (FND). FND were described thoroughly in a recent report [4]. Instead of representing structural damage to the central nervous system, FND arises due to disordered neurological function which is not explained or explainable by any other recognized neurological disorder. Symptoms and signs in these patients are real and nonvolitional (in contrast to what has been perceived in the past). These may be the typical responses of the body to the normal physiological processes occurring after immunization (i.e., pain or inflammation), and stress or fear of side effects may aggravate this symptomology. Well-characterized functional disorders in other systems include irritable bowel syndrome, vasovagal syncope, stress-associated dyspepsia, and stress-aggravated migraine headache [1].

A theory that may explain the occurrence of vaccine-associated stroke-like symptomology based on a mechanism involving complex regional pain syndrome and FND has been proposed in a prior report [5]. This review proposed that the peripheral inflammation or pain, in this case, after a vaccine injection, along with anxiety or excessive self-monitoring, intertwined with widespread central nervous system maladaptation. Such a physiologic response to acute pain is characterized by the redistribution of muscular activity, resulting in stiffening, limitation, and slow movement [6]. Normally temporary and under conscious control, this response may grow beyond conscious control and manifest as weakness. It was observed that most patients develop clinical symptoms of dysesthesia or numbness in the same limb as the immunization site. Further, the patients with stroke-like symptoms reported in our cohort had good prognoses; most patients recovered fully, and none had structural deficits.

Based on our findings, we encourage more comprehensive health education regarding this ISRR for both vaccine recipients and administering healthcare workers, who are responsible for effective clinical decision-making, meticulously monitoring patients' health, and reassuring patients that their symptoms are transient responses and that they will recover. These adverse events should not impede vaccination campaigns, especially during the pandemic. Appropriate follow-up investigation is also essential so as not to miss structural neurological deficits as well as to provide appropriate investigation and treatment when necessary.

The ISRR clusters evaluated in our study were reported when the worldwide vaccine phobia wave initially emerged due to widespread and frequently inaccurate information regarding the side effects of vaccination, which was highly prevalent on social media at the time. Following the first stroke-like events cluster reported to the Thai Ministry of Public Health in April 2021, highlighted by social media, many subsequent clusters were reported from several provinces around the country [7]. Vaccine constituents were the suspected culprits at the start of the vaccine phobia wave. Many patients were diagnosed with stroke and received thrombolytic or antiplatelet therapy because the normal brain CT scan and the clinical presentations could not confidently exclude stroke. These events quickly spread by social media led to vaccine hesitancy due to false belief of vaccine toxicity, causing many people to decide not to be vaccinated.

Following the appropriate investigation and a confirmation of the occurrence of ISRR, preparedness and management strategies were introduced on a nationwide scale. The number of reported cases was thus reduced substantially.

The first case report of stroke-like symptomology occurring after CoronaVac vaccination reported to the Ministry of Public Health was proposed to be an acute prolonged motor aura [8]. However, most later patients did not develop headaches. We also recorded that these adverse events were likely to be lateralized to the side of the injection. Therefore, we propose that the underlying mechanisms for vaccine-associated ISSR may be of peripheral origin (e.g., pain following vaccination) and be integrated with an autonomic response (i.e., a response

resembling a reflex sympathetic response, as in complex regional pain syndrome). Further, according to previous research, this symptomology may involve the central nervous system, as demonstrated by abnormalities in the postcentral gyrus and inferior parietal cortex, which are responsible for afferent information processing [9, 10]. This may explain the transient clinical weakness seen in some of our enrolled patients.

ISRR was previously perceived as psychogenic within the medical community. However, these symptoms are caused by functional physiologic changes rather than structural damage following the vaccination procedure. The same situation has been reported with many vaccines in the past [11–15]. Most of these were presented as non-epileptic seizures, fainting and dizziness mostly in the adolescent groups. In these reports, there were weaknesses in some cases, but no detailed pattern of weakness was recorded. In our report, all participants were adults (which may reflect the vaccination campaign firstly done in adults), and the pattern of weakness was similar to stroke (acute hemisensory deficit or hemiparesis). Most of our patients were female, which could be due to the initial target population of the nationwide vaccine campaign, which initially comprised HCWs who received preferential vaccination at the start of the vaccine campaign, most of whom were young females (e.g., nurses, therapists). Many underlying mechanisms have been proposed, including pain due to the vaccination process or inflammation occurring after vaccination, which may activate the peripheral nerves and the sympathetic nervous system, as explained in a previous description of the proposed complex regional pain syndrome mechanism [5].

Recently, two cases were reported wherein the patients developed a clinical level of stroke-like weakness after receiving an mRNA-based COVID-19 vaccine [3]. Another case report described hemiparesis on the same side at the injection site that lasted for 40 min. However, this patient developed hypoesthesia on the alternate side, and the neurological examination revealed midline splitting of the sensory deficit; this patient had normal brain CT and magnetic resonance imaging findings. The authors hypothesized that increased attention toward body signals and abnormal expectations regarding the symptoms of vaccination-induced injury might be responsible for this symptomology [16]. This report confirms that the process of incident vaccine-associated neurological disorders is more likely due to the vaccination process rather than specific vaccine constituents since these events have occurred within various vaccine platforms.

As COVID-19 mass vaccination has been widely implemented among high-risk populations with underlying diseases, we found that some confirmed stroke cases were simply temporally associated with vaccination within this study. This caveat provides precautionary information and informs the provision of appropriate, timely treatment (vs. empirically treating ISRR in the absence of differential diagnosis).

We acknowledge several limitations of this report. Specifically, this evaluation was based on a retrospective study within the secondary data, and we may have missed some critical information (especially with respect to outcome data). Moreover, we were unable to identify the factors associated with ISRR in this descriptive study and instead presented a descriptive evaluation only. Future studies should comprehensively evaluate risk factors and causality using multivariate modeling and a prospective design.

## Conclusions

To the best of our knowledge, our study reports the largest series regarding ISRR associated with neurological symptoms identified to date. Stroke-like symptoms are more likely to occur on the left side. During the vaccination campaign, the non-dominant arm was encouraged for the side of vaccination. However, that the symptoms occurred on the same side of the injection

was not ascertained as most cases had no record of the injection side in the medical record. We hypothesized that as most people are right-handed; thus most injections would mostly be on the left side and that the painful stimuli from the injection triggered and exaggerated self-attention. These would be the main mechanisms mediating the development of this transient symptom. A high degree of awareness concerning this symptomology is important to avoid over-investigation, which could result in unnecessary costs and complications. It is important to provide effective and comprehensive education regarding these events to the public as well as to the administering HCWs. These occurrences are real but transient and almost invariably present with excellent recovery prospects. This information and more comprehensive information obtained in future studies may reduce vaccine hesitancy and help ensure a successful nationwide vaccination program and inform interventions on a global scale.

## Supporting information

**S1 Data.**
(XLSX)

## Author Contributions

**Conceptualization:** Metha Apiwattanakul, Narupat Suanprasert, Arada Rojana-Udomsart, Thanes Termglinchan, Chaichana Sinthuwong, Tasanee Tantirittisak, Suchat Hanchaiphiboolkul, Kulkanya Chokephaibulkit.

**Data curation:** Metha Apiwattanakul, Narupat Suanprasert, Arada Rojana-Udomsart, Thanes Termglinchan, Chaichana Sinthuwong, Apinya Panjangampatthana, Jirapa Chimmanee, Kulkanya Chokephaibulkit.

**Formal analysis:** Metha Apiwattanakul, Pantep Angchaisuksiri, Suphot Srimahachota, Jurai Wongsawat, Somjit Stiudomkajorn, Sasisopin Kiertiburanakul, Chonnamet Techasaensiri, Wannada Laisuan, Weerawat Manosuthi, Pawinee Doungngern, Wereyarmarst Jaroenkunathum, Teeranart Jivapaisarnpong, Kulkanya Chokephaibulkit.

**Funding acquisition:** Metha Apiwattanakul.

**Investigation:** Metha Apiwattanakul.

**Methodology:** Metha Apiwattanakul.

**Project administration:** Metha Apiwattanakul.

**Supervision:** Tasanee Tantirittisak, Suchat Hanchaiphiboolkul, Kulkanya Chokephaibulkit.

**Writing – original draft:** Metha Apiwattanakul.

**Writing – review & editing:** Metha Apiwattanakul, Narupat Suanprasert, Arada Rojana-Udomsart, Thanes Termglinchan, Chaichana Sinthuwong, Pantep Angchaisuksiri, Suphot Srimahachota, Jurai Wongsawat, Somjit Stiudomkajorn, Sasisopin Kiertiburanakul, Chonnamet Techasaensiri, Wannada Laisuan, Weerawat Manosuthi, Pawinee Doungngern, Wereyarmarst Jaroenkunathum, Teeranart Jivapaisarnpong, Apinya Panjangampatthana, Jirapa Chimmanee, Kulkanya Chokephaibulkit.

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
