## [Decision Letter · Decision Letter 0]

12 Jun 2022

PONE-D-22-07310Good recovery of immunization stress-related responses presenting as cluster of stroke-like events following CoronaVac and ChAdOx1 vaccinationsPLOS ONE

Dear Dr. Apiwattanakul,

Thank you for submitting your manuscript to PLOS ONE. After careful consideration, we feel that it has merit but does not fully meet PLOS ONE’s publication criteria as it currently stands. Therefore, we invite you to submit a revised version of the manuscript that addresses the points raised during the review process.

The two reviewers addressed several major and minor concerns about your manuscript. Please revise your manuscript carefully.

We look forward to receiving your revised manuscript.

Kind regards,

Kenji Hashimoto, PhD

Section Editor

PLOS ONE

Journal Requirements:

2. As part of your revision, please complete and submit a copy of the Full ARRIVE 2.0 Guidelines checklist, a document that aims to improve experimental reporting and reproducibility of animal studies for purposes of post-publication data analysis and reproducibility: https://arriveguidelines.org/sites/arrive/files/Author%20Checklist%20-%20Full.pdf (PDF). Please include your completed checklist as a Supporting Information file. Note that if your paper is accepted for publication, this checklist will be published as part of your article

Reviewers' comments:

Reviewer's Responses to Questions

**Comments to the Author**

1. Is the manuscript technically sound, and do the data support the conclusions?

Reviewer #1: Yes

Reviewer #2: Yes

2. Has the statistical analysis been performed appropriately and rigorously? 

Reviewer #1: Yes

Reviewer #2: Yes

3. Have the authors made all data underlying the findings in their manuscript fully available?

Reviewer #1: Yes

Reviewer #2: Yes

4. Is the manuscript presented in an intelligible fashion and written in standard English?

Reviewer #1: Yes

Reviewer #2: Yes

5. Review Comments to the Author

Reviewer #1: Thank you for the opportunity to review this paper. It is a retrospective case notes review of patients who were diagnosed with stroke-like illnesses following COVID-19 vaccination. In my opinion this is an important topic and the authors have summarised perhaps the largest data on this topic so far (at least that I am aware of) and should be commended for that. Nevertheless I have several suggestions for improvement.

Introduction

I think the introduction was okay but there was only a single reference.

I think the introduction could have included some discussion of previous data from the COVID-19 pandemic and beyond which have indicated the possibility of (functional) neurological symptoms post vaccination (for references please see Table 2 in the following: https://jnnp.bmj.com/content/jnnp/92/11/1144.full.pdf).

I think the language needs tightening up in places to make it more precise, for example 'However, the relevant vaccine types were investigated and no problems were found' is very imprecise. What does 'no problems' mean.

The authors do elaborate on this in the Discussion but I think the introduction would also benefit from some information on what ISRRs are (as we argue in our previous piece, we feel that many ISRRs are better seen as functional neurological disorders: https://neuro.psychiatryonline.org/doi/full/10.1176/appi.neuropsych.21050116)

Methods

The methods need to be clarified in some parts in my opinion. I didn't get a good sense of how the authors accessed the AEFI reports. My guess is that they are employed by the institution which manages the AEFI reports, but this is not clear.

Related to this, I think the reader should be given a sense of when and how a clinician might submit an AEFI report. I would also like to be given a better sense of the information that allowed the authors to conclude this was ISRR/FND - were the patients examined and found to have positive clinical signs, etc?

Results

'This patient had clinical symptoms and laboratory data compatible with ISRR; the causes of this adverse event are currently being explored by a multidisciplinary team.' - if the clinical picture is consistent with ISRR, why is the team currently investigating?

I don't think it's a particular issue, but it's a bit unclear what BMI has to do with anything.

Discussion

I think the discussion makes some good points but needs significant work. The structure is not always logical and the authors don't make enough attempts to situate their findings within the extant literature.

As with the introduction, the discussion is poorly referenced. The authors make several assertive points in the first paragraph without a single reference. There are frequent other areas of the discussion which are lacking in references.

I think the sentence 'stroke-like syndromes occurring as clustered events have not been reported as adverse vaccine-associated events to date' is possibly true, however there have been several published reports of neurological symptoms following vaccinations which were diagnosed as FND/mass psychogenic illness which probably warrant mention (see references above).

I don't think the sentence 'ISRR is considered an alternative term for conversion disorder and/or dissociative neurological symptomology' is empirically true in terms of what the WHO believe (although I am in agreement with the authors if that is representative of their own opinion). As well as this, the notion of 'conversion disorder' is a bit outdated, with patients (at least in the UK and US) preferring functional neurological disorder, which is more aetiologically neutral.

'FND are not malingering and are rather neurological deficits that are not caused by structural lesions and are instead characterized by changes in the functionality of neurons and glia as well as physiologic changes occurring in specific brain regions.' - this sentence is pretty wordy but doesn't say very much - I'd recommend something like 'Instead of representing structural damage to the central nervous system, FND arises due to disordered neurological function which is not explained or explainable by any other recognised neurological disorder.'

I am intrigued by your suggestion that there may be likes between ISRR and CRPS however was slightly disappointed that this wasn't elaborated on in any meaningful way. What is the mechanism you allude to?

'reassuring patients that their symptoms are normal responses' - I get what the authors are saying and agree with the broad thrust, but it's a bit of a stretch in my opinion to suggest these are 'normal' responses.

'miss real structural neurological' - I'd remove 'real' here - it implies that ISRR/FND is not real.

'In Thailand, case clustering has been reported throughout the country' - what does this mean? Where is the reference?

'Moreover, many patients were diagnosed with stroke and received unnecessary thrombolytic or antiplatelet therapy despite normal findings on brain CT and/or MRI' - I get what the authors are saying here, and certainly agree with the idea that we should not be unnecessarily treating FND as stroke - but the presence of a normal CT in the acute phase does not mean that an ischaemic stroke is not presence and is therefore does not necessarily mean thrombolysis is inappropriate. FND should be diagnosed and distinguished from stroke in other ways, for example via positive clinical signs.

'These and similar effects and phenomena severely impede the efficacy and reach of vaccination campaigns.' How? Why?

'WHO consultation and preparedness guidance were also extremely helpful in facilitating this process' - again this is quite informal and imprecise.

The paragraph which begins at 221 is very long and seems to deal with multiple points within the same paragraph. I am also questioning the need to elaborate so much on the case report.

'However, in our study, most patients did not develop headaches' - so what does this mean?

'ISRR was initially perceived as psychogenic within the medical community. However, these symptoms are real and not iatrogenic.' - I don't like the term psychogenic but it is not mutually exclusive with 'real' and therefore I think these sentences need to be reworded. I also disagree that they are not iatrogenic - they arise, by definition, from the medical procedure of vaccination.

We finally get to discussion of other clusters of post-vaccination neurological symptoms from the literature in line 238, however there is very limited discussion/comparison.

The paragraph from 243 - 247 could be removed without altering the usefulness of the discussion.

'The reported data regarding female predominance could be due to the initial target population of the nationwide vaccine campaign, which initially comprised HCWs who received preferential vaccination at the start of the vaccine campaign, most of whom were young females (e.g., nurses, therapists).' - this is not a limitation.

'will undoubtedly reduce vaccine hesitancy' - are you sure it will undoubtedly do so?

Kind regards,

Dr Matt Butler

Reviewer #2: The authors present descriptive results of individuals in Thailand reporting stress-related responses after either the CoronaVac or ChAdOx1 COVID-19 vaccines during a 5 month period. Authors find most symptoms were sensory and reported on the left side of the body. Most of the individuals recovered well. The manuscript will be strengthened if the authors consider the following points.

1. In the Abstract, authors should make clear that this study takes place in Thailand. They should also mention the two vaccines under consideration in the Methods section of the Abstract.

2. lines 127-28: Authors state they performed chi-square tests comparing between the vaccine types, but none of these results are provided. Also, given the small numbers in the ChAdOx1 vaccine group, Fisher's exact test may be more appropriate if they do in fact compare the groups.

3. line 128: it is not clear why BMI was compared to a theoretical mean of 25 and what the goal of this one-sided test is.

4. line 146: what data were missing to remove 42 patients from those reported to have ISRR?

5. Were there any individuals who had ISRR after both doses of the vaccine in this study? Authors should clarify this.

6. How is the median time to recovery calculated if not everyone fully recovered (authors state in line 165 that only 74.8% of those receiving the CoronaVac vaccine fully recovered).

7. lines 171-172: this is where it looks like authors present results from the one-sample t-test (point 3 above), but the interpretation that ISRR is not associated with being overweight or obese is not accurate based on this test. The t-test will test whether the mean BMI differs from 25. If the null hypothesis is not rejected (p>0.05), as in this case, all one can say is that BMI is not significantly different from 25. This is not a test of whether ISRR is associated with being overweight or obese.

Minor points.

1. lines 58 and 155: the upper bound of the IQR (42) for age of onset for those receiving the CoronaVac does not match what is stated in Table 1 (40).

2. line 61: it is not clear how the percentage for those developing symptoms on the left side was calculated (53.8%) since Table 2 has 144 individuals with symptoms on the left side (144/221 is not 53.8% nor is 144/209).

3. line 63: authors report 90% of those receiving ChAdOx1 recovered well, but they later report (line 169) that all ChAdOx1 recipients had good outcomes; both "recovered well" and "good outcomes" are defined in the same way. This inconsistency should be fixed or clarified.

4. line 136: authors should clarify that there were other vaccines available to people, since the sum of the doses of CoronaVac and ChAdOx1 is about 1 million less than the total number of doses administered stated in line 135.

5. Authors should be consistent in how they write ChAdOx1, since sometimes it is written as ChAdOX1.

6. line 162: authors say that the left side of the body was the injection side, but according to Table 1, only 57/221 were known to have the injection on the left side.

7. line 163: authors say that 50 patients had the injection side known, but Table 1 has 61 with known injection side. Authors should correct that and any related percentages in the text.

8. line 164: change "characteristic" to "characteristics"

9. lines 275-276: authors say stroke-like symptoms are more likely to occur on the same side as the injection, but this is not supported by the data, since the majority of participants have unknown side of injection.

6. PLOS authors have the option to publish the peer review history of their article (what does this mean?). If published, this will include your full peer review and any attached files.

Reviewer #1: **Yes: **Dr Matt Butler

Reviewer #2: No

---

## [Author Response · Author response to Decision Letter 0]

16 Jul 2022

Dear Editor

We would like to thank the Editor and Reviewers for their valuable time and feedback on our manuscript and appreciate the opportunity to submit the revised manuscript. We would like to respond to reviewers’ comments point-by-point as following:

Reviewer 1

1. Comment:

Introduction

I think the introduction was okay but there was only a single reference.

I think the introduction could have included some discussion of previous data from the COVID-19 pandemic and beyond which have indicated the possibility of (functional) neurological symptoms post vaccination (for references please see Table 2 in the following: https://jnnp.bmj.com/content/jnnp/92/11/1144.full.pdf).

Response: We have added the references in the introduction as suggested. (line 77-84)

 Kim DD, Kung CS, Perez DL. Helping the Public Understand Adverse Events Associated With COVID-19 Vaccinations: Lessons Learned From Functional Neurological Disorder. JAMA Neurol. 2021;78(7):789-90.

 Butler M, Coebergh J, Safavi F, Carson A, Hallett M, Michael B, et al. Functional Neurological Disorder After SARS-CoV-2 Vaccines: Two Case Reports and Discussion of Potential Public Health Implications. J Neuropsychiatry Clin Neurosci.33(4):345-8.

2. Comment:

I think the language needs tightening up in places to make it more precise, for example 'However, the relevant vaccine types were investigated and no problems were found' is very imprecise. What does 'no problems' mean.

Response: We revised the sentence to: “However, the relevant vaccine types were investigated and found within regulated quality without identifiable toxins or contamination.” (line 98-100). 

3. Comment:

The authors do elaborate on this in the Discussion but I think the introduction would also benefit from some information on what ISRRs are (as we argue in our previous piece, we feel that many ISRRs are better seen as functional neurological disorders: https://neuro.psychiatryonline.org/doi/full/10.1176/appi.neuropsych.21050116)

Response: We added the following sentences to provide some information of ISRR in the first paragraph of Introduction: “These events are not caused by side effects due to vaccine components. Instead, they arise from the stress reaction to the vaccination process/cascade, which could be precipitated or potentiated by the pain or other common side effects of the injections. The symptoms may develop immediately (within 5 minutes) following vaccination, such as vasovagal reaction, or later, such as non-epileptic seizures, abnormal movements after COVID-19 vaccination [2] or motor and sensory symptoms which demonstrate positive neurological signs (Hoover’s sign) which were diagnosed of functional neurological disorders [3]” (line 77-84)

4. Comment: 

Methods The methods need to be clarified in some parts in my opinion. I didn't get a good sense of how the authors accessed the AEFI reports. My guess is that they are employed by the institution which manages the AEFI reports, but this is not clear.

Response: We have clarified this in the second paragraph of Method line 120-122 as following “We conducted a retrospective study using the data received from the National AEFI surveillance program (AEFI-DDC) and the copy of the medical records of the cases that were determined to be ISRR by the AEFI committee.”

5. Comment:

Related to this, I think the reader should be given a sense of when and how a clinician might submit an AEFI report. I would also like to be given a better sense of the information that allowed the authors to conclude this was ISRR/FND - were the patients examined and found to have positive clinical signs, etc?

Response: We clarified this in the first paragraph of Methods line 107-113 as following: “The AEFI report system for the routine vaccine in the National Immunization Program (NIP) has been established in every public hospital for over twenty years. The dedicated hospital staff reports any serious AE or AE of special interest or AE that occurred in the cluster through the national AEFI surveillance program (AEFI-DDC) hosted by the Department of Disease Control, Ministry of Public Health. A medical record and imaging scan copy would be submitted for systematic review by the AEFI committee.” 

The diagnosis of ISRR were further clarified in line 116-119 as following: “The primary physicians may primarily diagnose ISRR; however, all were confirmed by the AEFI committee composed of neurologists, internists, pediatricians, cardiologists, hematologists, and allergists. In some cases of doubt, the committee would seek further external expert neurologist consultation.”

6. Comment: 

Results

'This patient had clinical symptoms and laboratory data compatible with ISRR; the causes of this adverse event are currently being explored by a multidisciplinary team.' - if the clinical picture is consistent with ISRR, why is the team currently investigating?

Response: This case is the very first case of ISRR in a large cluster and received high attention by the social medias at that time. She recovered from the symptoms of ISRR but then had recurrent of the same symptoms without receiving any further vaccination, and still had mRS score of 3 in the last follow-up. The psychiatric problem was suspected. We added more details in the sentence line 186-189 as following: “the causes of the symptom recurrence and persistent high mRS score, which was inconsistency and likely to be functional deficits, are explored by a multidisciplinary team, including the psychiatrist.”

7. Comment: I don't think it's a particular issue, but it's a bit unclear what BMI has to do with anything.

Response: Low BMI is known to associate with higher risk of vasovagal reaction and could have higher risk of ISRR. Therefore, we included the information of BMI in our report. 

Reference: Yamada T, Yanagimoto S: Dose-Response Relationship between the Risk of Vasovagal Syncope and Body Mass Index or Systolic Blood Pressure in Young Adults Undergoing Blood Tests. Neuroepidemiology 2017;49:31-33. doi: 10.1159/000479698

8. Comment: Discussion

I think the discussion makes some good points but needs significant work. The structure is not always logical and the authors don't make enough attempts to situate their findings within the extant literature.

As with the introduction, the discussion is poorly referenced. The authors make several assertive points in the first paragraph without a single reference. There are frequent other areas of the discussion which are lacking in references.

I think the sentence 'stroke-like syndromes occurring as clustered events have not been reported as adverse vaccine-associated events to date' is possibly true, however there have been several published reports of neurological symptoms following vaccinations which were diagnosed as FND/mass psychogenic illness which probably warrant mention (see references above).

Response: We revised the Discussion and have added the appropriate references as suggested. (line 201-205)

9. Comment:

I don't think the sentence 'ISRR is considered an alternative term for conversion disorder and/or dissociative neurological symptomology' is empirically true in terms of what the WHO believe (although I am in agreement with the authors if that is representative of their own opinion). As well as this, the notion of 'conversion disorder' is a bit outdated, with patients (at least in the UK and US) preferring functional neurological disorder, which is more aetiologically neutral.

Response: We replaced “conversion disorder” to “functional neurological disorder” in Line 206

10. Comment:

'FND are not malingering and are rather neurological deficits that are not caused by structural lesions and are instead characterized by changes in the functionality of neurons and glia as well as physiologic changes occurring in specific brain regions.' - this sentence is pretty wordy but doesn't say very much - I'd recommend something like 'Instead of representing structural damage to the central nervous system, FND arises due to disordered neurological function which is not explained or explainable by any other recognized neurological disorder.' 

Response: We revised the sentence as suggested. (line 207-209)

11. Comment: I am intrigued by your suggestion that there may be likes between ISRR and CRPS however was slightly disappointed that this wasn't elaborated on in any meaningful way. What is the mechanism you allude to?

Response: We have explained the mechanism of CRPS and FND (ISRR) with reference in the line 218-223.

12. Comment: 'reassuring patients that their symptoms are normal responses' - I get what the authors are saying and agree with the broad thrust, but it's a bit of a stretch in my opinion to suggest these are 'normal' responses.

Response: We have changed “normal response” to “transient responses” which will be more precise. (line 231)

13. Comment: 'miss real structural neurological' - I'd remove 'real' here - it implies that ISRR/FND is not real.

Response: We removed “real” as you suggested. (line 233-234)

14. Comment: 'In Thailand, case clustering has been reported throughout the country' - what does this mean? Where is the reference?

Response:. We revised the sentence to: “Following the first stroke-like events cluster reported to the Thai Ministry of Public Health in April 2021, highlighted by social media, many subsequent clusters were reported from several provinces around the country [7].” (Line 239-241)

15 Comment: 'Moreover, many patients were diagnosed with stroke and received unnecessary thrombolytic or antiplatelet therapy despite normal findings on brain CT and/or MRI' - I get what the authors are saying here, and certainly agree with the idea that we should not be unnecessarily treating FND as stroke - but the presence of a normal CT in the acute phase does not mean that an ischaemic stroke is not presence and is therefore does not necessarily mean thrombolysis is inappropriate. FND should be diagnosed and distinguished from stroke in other ways, for example via positive clinical signs.

Response: We agree with the reviewer comment. We revised the sentences to: “Many patients were diagnosed with stroke and received thrombolytic or antiplatelet therapy because the normal brain CT scan and the clinical presentations could not confidently exclude stroke.” (line 242-244)

16. Comment:

'These and similar effects and phenomena severely impede the efficacy and reach of vaccination campaigns.' How? Why?

Response: We have clarified more on this sentence as “These events quickly spread by social media led to vaccine hesitancy due to false belief of vaccine toxicity, causing many people to decide not to be vaccinated.” (line 244-246)

17. Comment: 'WHO consultation and preparedness guidance were also extremely helpful in facilitating this process' - again this is quite informal and imprecise.

Response: We deleted this sentence.

18. Comment: The paragraph which begins at 221 is very long and seems to deal with multiple points within the same paragraph. I am also questioning the need to elaborate so much on the case report. 

'However, in our study, most patients did not develop headaches' - so what does this mean?

Response: We agree with the reviewer. We have shortened this paragraph. We revised the first sentence of this paragraph to: “The first case report of stroke-like symptomology occurring after CoronaVac vaccination reported to the Ministry of Public Health was proposed to be an acute prolonged motor aura [8]. However, most later patients did not develop headaches.” (line 250-252)

19. Comment: 'ISRR was initially perceived as psychogenic within the medical community. However, these symptoms are real and not iatrogenic.' - I don't like the term psychogenic but it is not mutually exclusive with 'real' and therefore I think these sentences need to be reworded. I also disagree that they are not iatrogenic - they arise, by definition, from the medical procedure of vaccination.

Response: We agree with the reviewer and revised the sentence to: “ISRR was previously perceived as psychogenic within the medical community. However, these symptoms are caused by functional physiologic changes rather than structural damage following the vaccination procedure.” (line 261-263)

20. Comment: We finally get to discussion of other clusters of post-vaccination neurological symptoms from the literature in line 238, however there is very limited discussion/comparison.

Response: We have added more references and discussion. (line 263-266)

21. Comment: The paragraph from 243 - 247 could be removed without altering the usefulness of the discussion.

Response: We have removed this paragraph as suggested.

22. Comment: 'The reported data regarding female predominance could be due to the initial target population of the nationwide vaccine campaign, which initially comprised HCWs who received preferential vaccination at the start of the vaccine campaign, most of whom were young females (e.g., nurses, therapists).' - this is not a limitation.

Response: We moved these statements to line 268-272, not limitation.

23. Comment: 'will undoubtedly reduce vaccine hesitancy' - are you sure it will undoubtedly do so?

Response: We have revised to “may reduce vaccine hesitancy”, line 313-314

Reviewer #2: The authors present descriptive results of individuals in Thailand reporting stress-related responses after either the CoronaVac or ChAdOx1 COVID-19 vaccines during a 5 month period. Authors find most symptoms were sensory and reported on the left side of the body. Most of the individuals recovered well. The manuscript will be strengthened if the authors consider the following points.

1. In the Abstract, authors should make clear that this study takes place in Thailand. They should also mention the two vaccines under consideration in the Methods section of the Abstract.

Response: We have add “in Thailand” as you suggested (line 46) and also mention the two vaccines in Methods section. (line 48).

2. lines 127-28: Authors state they performed chi-square tests comparing between the vaccine types, but none of these results are provided. Also, given the small numbers in the ChAdOx1 vaccine group, Fisher's exact test may be more appropriate if they do in fact compare the groups.

Response: As we have no data using Chi-square nor Fisher’s exact test, so we removed this sentence.

3. line 128: it is not clear why BMI was compared to a theoretical mean of 25 and what the goal of this one-sided test is.

Response: The low BMI is associated with higher risk of vasovagal reaction and could have higher risk of ISRR. But we agree that this cannot be compared to the theoretical mean of 25 so we deleted this statistical test and the descriptive data of mean BMI was remained. 

4. line 146: what data were missing to remove 42 patients from those reported to have ISRR?

Response: The 42 patients were removed from the analysis due to lack of important information as following: time from vaccination to neurological symptoms onset 14 cases), time to recovery (42 case), age (29 cases), follow-up functional status (42 cases). 

5. Were there any individuals who had ISRR after both doses of the vaccine in this study? Authors should clarify this.

Response: None of patients developed ISRR after both doses of vaccination. Most of the patients did not receive the same vaccine in the subsequent dose, and some refused to get the second dose of any vaccine. We have added this sentence in line 191.

6. How is the median time to recovery calculated if not everyone fully recovered (authors state in line 165 that only 74.8% of those receiving the CoronaVac vaccine fully recovered).

Response: We calculated using the most recent follow up dated as available in the medical records or July 31st, 2021, which ever was reached first.

7. lines 171-172: this is where it looks like authors present results from the one-sample t-test (point 3 above), but the interpretation that ISRR is not associated with being overweight or obese is not accurate based on this test. The t-test will test whether the mean BMI differs from 25. If the null hypothesis is not rejected (p>0.05), as in this case, all one can say is that BMI is not significantly different from 25. This is not a test of whether ISRR is associated with being overweight or obese.

Response: We presented only the descriptive analysis of BMI, and deleted the sentence that indicated the association.

Minor points.

1. lines 58 and 155: the upper bound of the IQR (42) for age of onset for those receiving the CoronaVac does not match what is stated in Table 1 (40).

Response: We corrected the data in Table 1.

2. line 61: it is not clear how the percentage for those developing symptoms on the left side was calculated (53.8%) since Table 2 has 144 individuals with symptoms on the left side (144/221 is not 53.8% nor is 144/209).

Response: We have corrected the error to 68.9% (line 64).

3. line 63: authors report 90% of those receiving ChAdOx1 recovered well, but they later report (line 169) that all ChAdOx1 recipients had good outcomes; both "recovered well" and "good outcomes" are defined in the same way. This inconsistency should be fixed or clarified.

Response: We revised the wording to be “all had good outcome” consistently (line 65-66 and 189).

4. line 136: authors should clarify that there were other vaccines available to people, since the sum of the doses of CoronaVac and ChAdOx1 is about 1 million less than the total number of doses administered stated in line 135.

Response: We have added the information of other vaccine (BBIBP-CorV) in line 154.

5. Authors should be consistent in how they write ChAdOx1, since sometimes it is written as ChAdOX1.

Response: We corrected all the typo errors (line 156).

6. line 162: authors say that the left side of the body was the injection side, but according to Table 1, only 57/221 were known to have the injection on the left side.

Response: We revised by deleting “injection side” (line 179).

7. line 163: authors say that 50 patients had the injection side known, but Table 1 has 61 with known injection side. Authors should correct that and any related percentages in the text.

Response: The data in the Table is correct. We revised the number in the line 181-182.

8. line 164: change "characteristic" to "characteristics"

Response: We corrected the error (line 183).

9. lines 275-276: authors say stroke-like symptoms are more likely to occur on the same side as the injection, but this is not supported by the data, since the majority of participants have unknown side of injection.

Response: We have clarified more on the line 301-308 as “Stroke-like symptoms are more likely to occur on the left side. During the vaccination campaign, the non-dominant arm was encouraged for the side of vaccination. However, that the symptoms occurred on the same side of the injection was not ascertained as most cases had no record of the injection side in the medical record. We hypothesized that as most people are right-handed; thus most injections would mostly be on the left side and that the painful stimuli from the injection triggered and exaggerated self-attention. These would be the main mechanisms mediating the development of this transient symptom.” 

Sincerely,

Metha Apiwattanakul, MD

Corresponding Author

---

## [Decision Letter · Decision Letter 1]

9 Aug 2022

Good recovery of immunization stress-related responses presenting as a cluster of stroke-like events following CoronaVac and ChAdOx1 vaccinations

PONE-D-22-07310R1

Dear Dr. Apiwattanakul,

We’re pleased to inform you that your manuscript has been judged scientifically suitable for publication and will be formally accepted for publication once it meets all outstanding technical requirements.

Kind regards,

Kenji Hashimoto, PhD

Section Editor

PLOS ONE

Additional Editor Comments (optional):

Reviewers' comments:

Reviewer's Responses to Questions

**Comments to the Author**

1. If the authors have adequately addressed your comments raised in a previous round of review and you feel that this manuscript is now acceptable for publication, you may indicate that here to bypass the “Comments to the Author” section, enter your conflict of interest statement in the “Confidential to Editor” section, and submit your "Accept" recommendation.

Reviewer #1: All comments have been addressed

Reviewer #2: All comments have been addressed

2. Is the manuscript technically sound, and do the data support the conclusions?

Reviewer #1: Yes

Reviewer #2: (No Response)

3. Has the statistical analysis been performed appropriately and rigorously? 

Reviewer #1: Yes

Reviewer #2: (No Response)

4. Have the authors made all data underlying the findings in their manuscript fully available?

Reviewer #1: Yes

Reviewer #2: (No Response)

5. Is the manuscript presented in an intelligible fashion and written in standard English?

Reviewer #1: Yes

Reviewer #2: (No Response)

6. Review Comments to the Author

Reviewer #1: Thanks to the authors who've done an excellent job responding to the comments.

The paper is now a good read and covers an important topic.

No further suggestions.

Reviewer #2: (No Response)

7. PLOS authors have the option to publish the peer review history of their article (what does this mean?). If published, this will include your full peer review and any attached files.

Reviewer #1: **Yes: **Dr Matt Butler

Reviewer #2: No

---

## [Editor Report · Acceptance letter]

16 Aug 2022

PONE-D-22-07310R1 

Good recovery of immunization stress-related responses presenting as a cluster of stroke-like events following CoronaVac and ChAdOx1 vaccinations 

Dear Dr. Apiwattanakul:

I'm pleased to inform you that your manuscript has been deemed suitable for publication in PLOS ONE. Congratulations! Your manuscript is now with our production department. 

Kind regards, 

on behalf of

Prof. Kenji Hashimoto 

Section Editor

PLOS ONE